# Chemical Characterization, Antioxidant Activity, and Cytotoxity of Wild-Growing and In Vitro Cultivated *Rindera umbellata* (Waldst. and Kit.) Bunge

Slađana Todorović [1,*], Marija Perić [1,2], Biljana Nikolić [3], Boris Mandić [4], Stefana Cvetković [3], Milica Bogdanović [1] and Suzana Živković [1]

1   Institute for Biological Research "Siniša Stanković"-National Institute of Republic of Serbia, University of Belgrade, Bulevar Despota Stefana 142, 11060 Belgrade, Serbia
2   Sanitary Medical School of Applied Sciences VISAN, Tošin bunar 7a, 11000 Belgrade, Serbia
3   University of Belgrade, Faculty of Biology, Studentski trg 16, 11000 Belgrade, Serbia
4   University of Belgrade, Faculty of Chemistry, Studentski trg 12-16, 11000 Belgrade, Serbia
*   Correspondence: slatod@ibiss.bg.ac.rs

**Abstract:** The aim of this study was to comparatively analyze chemical composition and biological activity of wild- and in vitro grown *Rindera umbellata*. Explants were cultivated on 0.003–0.3 M sucrose, fructose, or glucose. HPLC-DAD for quantifying rosmarinic (RA) and lithospermic B (LAB) acids and GC-MS/FID for qualitative pyrrolizidine alkaloids (PAs) detection were used. Antioxidant activity (DPPH and ABTS assays) and cytotoxicity (MTT test) were monitored. Identified PAs were 7-angeloyl heliotridane, lindelofine, 7-angeloyl heliotridine, 7-angeloyl-9-(+)-trachelanthylheliotridine, punctanecine, and heliosupine, with higher variability reported in wild-growing samples. Total phenolic contents (TPCs) were comparable in wild-growing and in vitro samples, but total flavonoid (TFC) and RA levels were multifold higher in in vitro samples. Notably, high concentration of LAB was detected in wild-growing roots. Amounts of 0.3 M and 0.1 M of sucrose were optimal for TFC and RA production, while maximal antioxidant activity was monitored in plants grown on 0.3 M sucrose. The MTT test indicated colorectal HT-29 as more sensitive than A549 lung adenocarcinoma and normal MRC-5 cells, showing selective sensitivity to wild-growing and 0.3 M sucrose samples. In conclusion, PAs in vitro, as well as TPC, TFC, RA, and LAB in both growing conditions were detected for the first time in *R. umbellata*.

**Keywords:** ex situ conservation; rosmarinic acid; lithospermic acid B; pyrrolizidine alkaloids; biological activity

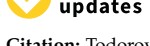



## 1. Introduction

*Rindera umbellata* (Waldst. and Kit.) Bunge, also known as *Mattia umbellata* (Waldst. and Kit.) Schult. belongs to the Boraginaceae family and its distribution is limited to sandy habitats of southeast Europe, from the Central Balkan Peninsula to southwest Ukraine. In Serbia, *R. umbellata* is a rare and critically endangered species, found only in Deliblato and Kladovo Sands. Due to its rarity and extinction risk, *R. umbellata* is under National State Protection Legislation [1]. Propagation of *R. umbellata* in nature is limited and laborious due to fungal infections and insufficient seed maturity [2]. To preserve the existing gene fond and provide plant material for further molecular and phytochemical analysis, establishment of adequate methods of conservation is necessary. One of the ex situ conservation strategies of rare and endangered plant species is cultivation of plants in tissue culture. The method of in vitro cultivation provides production of healthy plant material suitable for environmental recultivation and further phytochemical research. Moreover, in vitro cultivated plants can be considered as a valuable source of biologically active compounds that can be applied in different fields, i.e., as pharmaceuticals, flavors

and fragrances, food additives, agrochemicals, and biopesticides [3,4]. However, in vitro cultivation requires some specificity, such as the necessity to add carbohydrates in a culture media. This can be addressed to limited photosynthetic activity, due to low light intensity and restricted $CO_2$ amounts in a closed system used for propagation [5]. Moreover, concentration of carbohydrates in culture media does not affect yield of plant material only, but also the level of secondary metabolites production [6]. Carbohydrate types and concentrations in culture media induce metabolic changes and, consequently, the adequate conditions stimulating production of secondary metabolites can be achieved [7]. This is especially important since the abundant secondary metabolites are embedded with numerous biological activities and could be used as a raw material for different pharmaceutical products [3].

Plant phenolic compounds are an important group of secondary metabolites in which production strongly depends on carbohydrate content [8]. Phenolics are a chemically heterogeneous group of compounds broadly distributed in the plant kingdom, showing a great diversity of structure and roles for the plants themselves [9]. Numerous literature data indicate that phenolic compounds possess strong antioxidant potential and could remove free radicals and promote chelation of redox active metal ions, consequently preventing lipid peroxidation and Fenton reaction [10]. Another important group of plant secondary metabolites are alkaloids, chemical entities classified into several subgroups, with a primary function in protecting against herbivores and microbial infections [11]. Both phenolic compounds and alkaloids could affect human health due to diverse biological activities [12,13].

Among phenolic compounds, rosmarinic acid (RA), an ester of caffeic and 3,4-dihydroxyphenyllactic acid, and a dimer of rosmarinic acid denoted as lithospermic acid B (LAB, also known as salvianolic acid B), are well known for variable biological activities. Literature review showed different effects of RA, such as analgesic, anticancer, antiallergic, neuroreparative, cardioprotective, and antidiabetic; many of these activities can be attributed, at least partially, to its potent antioxidative and anti-inflammatory properties [14]. Similarly, LAB is well known to induce antioxidant, anti-inflammatory, anticancer, antihypertension, and cardioprotective effects [15]. Both substances are featured with notable anti-thrombosis and antiviral (anti-HIV) properties, and could be used as aldose reductase inhibitors, being active as eye protectors in diabetic patients [16].

In a previous study conducted by Perić et al. [17], *R. umbellata* was introduced in in vitro culture and growth conditions for its propagation were optimized. Furthermore, its acclimatization to greenhouse and field was successfully established. Some of our previous researches detected pyrrolizidine alkaloids (PAs) and fatty acids of wild-growing *R. umbellata* [18,19]. However, literature review revealed no publications dedicated to phytochemical composition of in vitro grown *R. umbellata*. Considering the abovementioned facts, the subject of this study was to comparatively analyze the chemical composition of methanol extracts of wild-harvested and in vitro cultivated *R. umbellata*, as well as to investigate the effect of selected carbohydrates, namely, sucrose, glucose, and fructose, on the production of phenolic compounds. Furthermore, selected extracts were examined for the antioxidant capacity and cytotoxic effect against lung and colorectal adenocarcinoma cells (A549 and HT-29 lines, respectively). Our investigation has been conducted to optimize culture conditions for providing a source of valuable biologically active compounds. In addition, *R. umbellata* in vitro culturing could diminish its uncontrolled harvesting from nature, contributing to protection and ex situ conservation of this rare and endangered species.

## 2. Materials and Methods

### 2.1. Wild-Growing Plant Material

*R. umbellata* plants were collected from a population distributed in Deliblato Sands, Serbia, at Latitude N 44°57′58′′ and Longitude E 21°1′48′′, in accordance with permission for the collection of strictly protected and protected plant species for scientific research purposes No. 353-01-184/2011-03. Collected plant material was dried at room temperature.

Different parts of air-dried plants (flowers, stems, leaves, and roots) were used for the extracts preparation (Figure 1A,B).

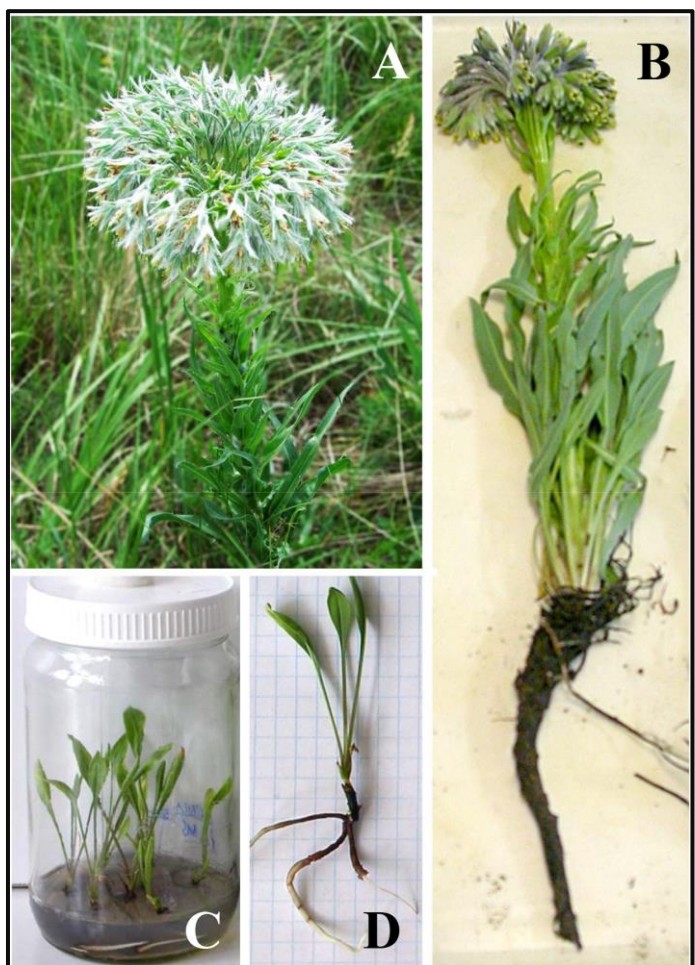

**Figure 1.** *Rindera umbellata* from the natural habitat (**A**,**B**) and grown in vitro (**C**,**D**).

### 2.2. In Vitro Culture Establishment

In vitro culture was established as described earlier [17] via embryos excised from immature seeds. Stems, 1 cm long, were used as primary explants. They were provided from adventitious buds propagated on basal medium (BM) containing 3% sucrose, 0.7% agar, and MS salts and vitamins [20] with addition of 1 mg $L^{-1}$ 6-benzylaminopurine (BAP) and 0.5 mg $L^{-1}$ indole-3-acetic acid (IAA). Explants were cultivated on media without growth regulators containing different carbohydrate sources (i.e., sucrose, fructose, or glucose, ranging from 0.003 to 0.3 M). After six weeks, dry aerial parts (rosette leaves) were used for the extraction (Figure 1C,D).

### 2.3. Preparation of Plant Extracts

Dry plant samples (0.1 g each) were powdered in liquid nitrogen and extracted at room temperature using 80% methanol (1:10, *w:v*). After centrifugation for 20 min at 15,000× *g*, supernatants were collected and filtered through 0.45 μm PVDF filters (Econofilter, Agilent Technologies, Santa Clara, CA, USA). For cytotoxicity evaluation they were additionally evaporated at room temperature and stored at +4 °C until use. To analyze the presence of PAs in the extracts of wild- and in vitro grown plant material, the samples were prepared in accordance with Mandić et al. [21].

### 2.4. Total Phenolic and Flavonoid Content Determination

Total phenolic content (TPC) and total flavonoid content (TFC) of plant methanol extracts were determined as reported earlier [22]. Briefly, for TPC estimation the extracts were mixed with 5% sodium carbonate and incubated for 5 min at room temperature. The extracts were then oxidized with 50% Folin—Ciocalteu reagent (Fermentas International Inc., Burlington, ON, Canada) and subsequently incubated for one hour at room temperature in a dark condition. The absorbance of the resulting mixture was measured at 724 nm using HP Agilent 8453 Spectrophotometer (Agilent Technologies, Palo Alto, CA, USA). Quantification was based on the standard curve of gallic acid. Results were expressed as milligram of gallic acid equivalent per g of dry weight (mg GAE $g^{-1}$ DW).

An aluminum chloride complex-forming assay (aluminum chloride colorimetric method) was used to determine TFC. Plant extracts were mixed with 5% $KNO_2$ and incubated at room temperature. After addition of 10% $AlCl_3$, a yellow complex solution was formed, which then turned immediately to red after addition of 1 M NaOH. The absorbance of the mixture was measured at 510 nm. TFC content in the extract was calculated from the standard curve based on rutin, and the results were expressed as milligram of rutin equivalent per g of dry weight (mg RE $g^{-1}$ DW).

### 2.5. HPLC-DAD Analysis of Phenolic Compounds

Chromatographic analysis of methanol extracts was performed on an Agilent HPLC system, model 1100, with DAD, using Hypersil column BDS-C18 (5 μm), 125 mm × 2 mm I.D (Phenomenex, Torrance, CA, USA). The samples were filtered through 0.2 μm cellulose filters (Agilent technologies, Santa Clara, CA, USA) prior to analysis. The mobile phase contained 0.1% phosphoric acid and acetonitrile (Acros Organics, Geel, Belgium). Phosphoric acid (A) and acetonitrile (B) were applied in elution gradient as follows: 7.5% B (0.00 min), 20% B (20.00 min), 25% B (25.00 min), and 7.5% B (30.00 min), with the flow rate of 0.500 mL $min^{-1}$. Chromatograms were monitored at different wavelengths (210 nm, 266 nm, 310 nm, and 326 nm). Quantification was performed using a standard curve prepared with five different concentrations of standard mixtures, containing coniferyl alcohol, rutin hydrate, gallic, chlorogenic, caffeic, *p*-coumaric, ferulic, and rosmarinic acid (Sigma-Aldrich, Oakville, ON, Canada). Chromatogram analysis was conducted by HP Chemstation chromatographic software (Palo Alto, CA, USA).

### 2.6. GC-MS/FID Analysis of Pyrrolizidine Alkaloids

Qualitative analysis GC-MS analyses were performed on an Agilent 7890A GC system equipped with a 5975C inert XL EI/CI MSD and a FID detector connected by capillary flow technology through a 2-way splitter with make-up gas. An HP-5 MS capillary column (Agilent Technologies, Santa Clara, CA, USA, 25 mm i.d., 30 m length, 0.25 μm film thickness) was used. Samples were injected in splitless mode. The injection volume was 1 μL and 3 μL for the extracts obtained from wild- and in vitro grown plants, respectively, while the injector temperature was 250 °C. The carrier gas (He) flow rate was 1.1 mL $min^{-1}$, whereas the column temperature was programmed linearly in a range of 100–325 °C at a rate of 10 °C $min^{-1}$. The transfer line temperature was 320 °C. The FID detector temperature was 300 °C. EI mass spectra (70 eV) were acquired in an $m/z$ range of 45–450, and the ion source temperature was 230 °C.

### 2.7. Determination of Antioxidant Activity

Free radical scavenging activity of the selected extracts was measured using DPPH and ABTS antioxidant capacity assays, as previously described by Šiler at al. [22]. The results are expressed using the $IC_{50}$ value, defined as the concentration of antioxidant that causes a 50% decrease in the DPPH or ABTS radical absorbance. RA was used as the reference compound. Results are expressed as mM of RAE per g of DW (mM RAE $g^{-1}$ DW). All the analyses were run in triplicate and mean values were calculated.

### 2.8. Cytotoxicity Assay

The human cell lines used in the cytotoxicity assay were human lung adenocarcinoma epithelial cells A549 (ATCC CCL-185), human colorectal adenocarcinoma cells HT-29 (ATCC HTB-38), and normal human fetal lung fibroblasts MRC-5 (ECACC 84101801). Cells were cultured in DMEM supplemented with 10% fetal bovine serum and 1% penicillin/streptomycin mixture (all purchased from Sigma-Aldrich, Steinheim, Germany). Cell lines were maintained in an incubator at 37 °C with 5.0% $CO_2$ and 100% humidity. The cells were sub-cultured at 90% confluence, twice each week, using 0.1% trypsin. The 3-(4,5-dimethylthiazol-2-yl)-2,5-diphenyltetrazolium bromide (MTT) was used as an indicator of metabolic activity/cell viability.

The cytotoxic effect of the selected extracts was measured by MTT assay [23]. Briefly, cells were inoculated into 96-well plates at a density $2 \times 10^4$ cells per well. After 24 h of pre-incubation at 37 °C in 5% $CO_2$, used to provide a cell monolayer, cells were washed with phosphate buffer saline (PBS). After that, fresh medium with tested concentrations of the extracts (within the range 0.125–4 mg mL$^{-1}$) was added and the treatment lasted for the following 24 h at 37 °C in 5% $CO_2$. After treatment, MTT solution (0.5 mg mL$^{-1}$) was added and an additional 3 h of incubation was provided in order to form formazan crystals. They were then dissolved in DMSO and absorbance was measured at 570 nm, indicating cell viability. For each test substance, three independent experiments with six replicates were performed. Furthermore, cytotoxicity data obtained on different cell lines were used to calculate selectivity index (SI), as previously described by Badisa et al. [24]. It was calculated according to the equation SI = IC$_{50}$ for normal cells/IC$_{50}$ for cancer cells. SI > 1 indicates higher toxicity against cancer cells.

### 2.9. Statistical Analysis

GraphPad Prism 6.01 software (Software, Inc., San Diego, CA, USA) was used and one-way ANOVA with Dunnet's and Tukey's post hoc tests was applied to test statistical significance at $p < 0.05$ level.

## 3. Results

### 3.1. Phenolic Compounds Content

Content of phenolic compounds was analyzed by determining the quantity of TPC and TFC, as well as the contribution of RA and LAB, as the dominant constituents (Figure 2).

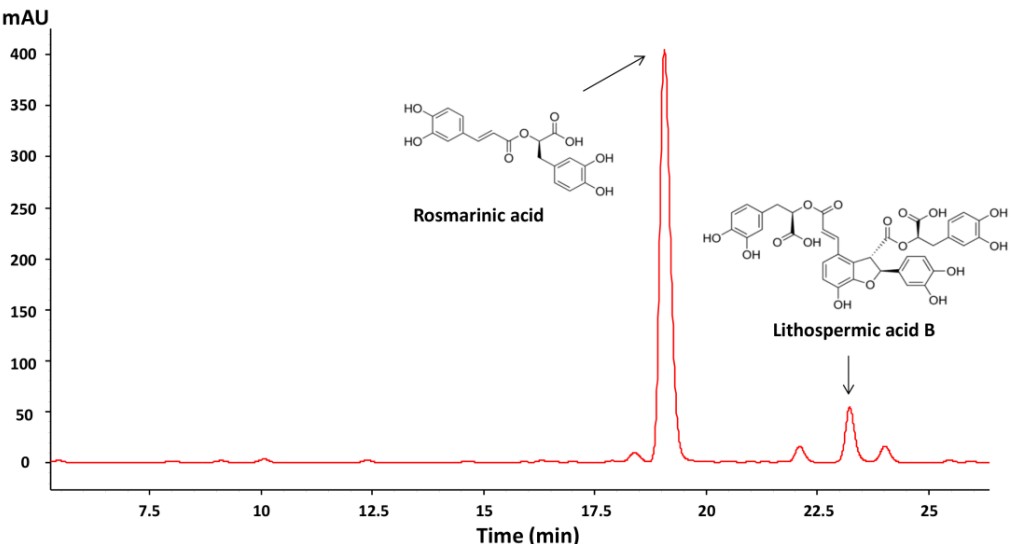

**Figure 2.** HPLC-DAD chromatogram of the methanol extract of *Rindera umbellata* from the natural habitat. Detection was performed at 326 nm. Major peaks corresponding to rosmarinic acid (Rt = 19.05 min) and lithospermic acid B (Rt = 23.25 min) are labeled.

The results revealed that quantities of all measured compounds varied between growth conditions and tested plant parts. While for wild-growing plants the content depended on plant part used for extract preparation (Table 1), for in vitro cultivated plants it was dependent on the type and concentration of carbohydrates used in cultivation media (Table 2). Concerning the wild-growing plants, the highest TPC (32.37 mg GAE g$^{-1}$ DW) was detected in the leaves, with 0.86% consisting of RA (0.28 mg g$^{-1}$ DW) and 0.12% consisting of LAB (0.04 mg RAE g$^{-1}$ DW). In their roots, total phenols were detected in the amount of 26.93 mg GAE g$^{-1}$ DW, with the highest proportion of LAB (4.15%, i.e., 1.12 mg RAE g$^{-1}$ DW). Moreover, roots contained higher amounts of LAB compared with leaves, while it was not detected in other tested plant parts (flowers and stem). In addition, there was a significant participation of flavonoids in TPC, with the highest content detected in leaves (22.87 mg RE g$^{-1}$ DW).

**Table 1.** Phenolic content in different parts of wild growing *R. umbellata* plants. Results represent the mean values $\pm$ SE of the three repeated measurements. Different letters indicate statistical difference at $p \leq 0.05$ level of significance. TPC—total phenolic content; TFC—total flavonoid content; RA—rosmarinic acid; LAB—lythospermic acid B.

| Plant Parts | TPC (mg GAE g$^{-1}$ DW) | TFC (mg RE g$^{-1}$ DW) | RA (mg g$^{-1}$ DW) | LAB (mg RAE g$^{-1}$ DW) |
|---|---|---|---|---|
| Leaves | 32.37 $\pm$ 0.15 *c* | 22.87 $\pm$ 2.12 *c* | 0.28 $\pm$ 0.04 *b* | 0.04 $\pm$ 0.02 *a* |
| Flowers | 25.31 $\pm$ 1.70 *b* | 13.41 $\pm$ 2.31 *b* | 0.24 $\pm$ 0.04 *b* | Nd |
| Roots | 26.93 $\pm$ 0.22 *b* | 19.10 $\pm$ 3.15 *bc* | 0.06 $\pm$ 0.01 *a* | 1.12 $\pm$ 0.21 *b* |
| Stems | 7.26 $\pm$ 1.17 *a* | 0.81 $\pm$ 0.44 *a* | 0.02 $\pm$ 0.01 *a* | Nd |

Nd—not determined.

**Table 2.** Phenolic content in leaves of *R. umbellata* plants grown in vitro using different carbohydrate sources. Results represent the mean values $\pm$ SE of the three repeated experiments. For each sugar and phenolic compound, the values with the different letters are significantly different at the $p \leq 0.05$ level. TPC—total phenolic content; TFC—total flavonoid content; RA—rosmarinic acid; LAB—lythospermic acid B.

| | Carbohydrate Concentration (M) | TPC (mg GAE g$^{-1}$ DW) | TFC (mg RE g$^{-1}$ DW) | RA (mg g$^{-1}$ DW) | LAB (mg RAE g$^{-1}$ DW) |
|---|---|---|---|---|---|
| Sucrose | 0.003 | 18.63 $\pm$ 1.78 *a* | 15.37 $\pm$ 2.86 *b* | 0.43 $\pm$ 0.04 *ab* | 0.05 $\pm$ 0.02 *a* |
| | 0.01 | 29.37 $\pm$ 0.96 *b* | 8.37 $\pm$ 1.48 *a* | 0.22 $\pm$ 0.03 *a* | 0.01 $\pm$ 0.01 *a* |
| | 0.03 | 29.58 $\pm$ 1.11 *b* | 16.65 $\pm$ 0.38 *bc* | 0.42 $\pm$ 0.01 *ab* | 0.01 $\pm$ 0.01 *a* |
| | 0.06 | 26.08 $\pm$ 1.86 *b* | 17.76 $\pm$ 2.29 *bc* | 0.46 $\pm$ 0.04 *ab* | 0.07 $\pm$ 0.01 *a* |
| | 0.1 | 27.87 $\pm$ 1.33 *b* | 22.15 $\pm$ 1.33 *c* | 0.71 $\pm$ 0.11 *b* | 0.10 $\pm$ 0.02 *a* |
| | 0.3 | 27.95 $\pm$ 1.40 *b* | 61.97 $\pm$ 1.49 *d* | 2.52 $\pm$ 0.17 *c* | 0.11 $\pm$ 0.11 *a* |
| Glucose | 0.003 | 13.92 $\pm$ 0.01 *a* | 10.83 $\pm$ 1.56 *abc* | 0.23 $\pm$ 0.06 *a* | 0.04 $\pm$ 0.02 *a* |
| | 0.01 | 14.43 $\pm$ 0.21 *a* | 8.23 $\pm$ 0.98 *ab* | 0.18 $\pm$ 0.01 *a* | 0.01 $\pm$ 0.01 *a* |
| | 0.03 | 18.02 $\pm$ 0.46 *b* | 13.55 $\pm$ 2.61 *bc* | 0.30 $\pm$ 0.06 *ab* | 0.02 $\pm$ 0.02 *a* |
| | 0.06 | 21.45 $\pm$ 0.20 *c* | 14.40 $\pm$ 2.56 *c* | 0.40 $\pm$ 0.06 *b* | 0.02 $\pm$ 0.01 *a* |
| | 0.1 | 20.26 $\pm$ 0.01 *c* | 12.56 $\pm$ 0.66 *bc* | 0.28 $\pm$ 0.05 *ab* | 0.00 $\pm$ 0.00 *a* |
| | 0.3 | 26.74 $\pm$ 0.97 *d* | 6.60 $\pm$ 1.69 *a* | 0.23 $\pm$ 0.06 *a* | 0.00 $\pm$ 0.00 *a* |
| Fructose | 0.003 | 17.95 $\pm$ 0.17 *b* | 14.10 $\pm$ 0.32 *ab* | 0.23 $\pm$ 0.01 *a* | 0.04 $\pm$ 0.15 *a* |
| | 0.01 | 19.25 $\pm$ 0.86 *b* | 11.26 $\pm$ 1.46 *a* | 0.26 $\pm$ 0.04 *a* | 0.06 $\pm$ 0.02 *ab* |
| | 0.03 | 24.50 $\pm$ 0.97 *c* | 10.68 $\pm$ 0.32 *a* | 0.24 $\pm$ 0.04 *a* | 0.05 $\pm$ 0.01 *a* |
| | 0.06 | 20.63 $\pm$ 1.53 *bc* | 14.94 $\pm$ 3.08 *ab* | 0.34 $\pm$ 0.08 *a* | 0.07 $\pm$ 0.02 *b* |
| | 0.1 | 13.75 $\pm$ 1.42 *a* | 18.43 $\pm$ 1.99 *b* | 0.46 $\pm$ 0.10 *a* | 0.08 $\pm$ 0.05 *b* |
| | 0.3 | 11.24 $\pm$ 1.18 *a* | 31.21 $\pm$ 2.55 *c* | 1.51 $\pm$ 0.22 *b* | 0.05 $\pm$ 0.05 *a* |

On the other hand, TPC of in vitro grown plants was measured only in rosette leaves, considering that root development of this species in culture is a slow process resulting in low biomass production (data not shown). The effect of different carbohydrates concentrations in the growth media was monitored for three commonly used carbohydrates, namely, sucrose, glucose, and fructose. The highest TPC was detected in leaves of *R. um-*

*bellata* grown on media containing 0.03 M sucrose and fructose, as well as 0.3 M glucose (29.58 mg GAE g$^{-1}$ DW, 24.50 mg GAE g$^{-1}$ DW, and 26.74 mg GAE g$^{-1}$ DW, respectively, Table 2).

For TFC and RA, an increased amount was observed only at 0.3 M sucrose and 0.3 M fructose (61.97 and 31.21 mg RE g$^{-1}$ DW of TFC, and 2.52 and 1.51 mg g$^{-1}$ DW of RA, respectively), while LAB was detected only in trace amounts, with the slightly higher content detected on media containing 0.1 M and 0.3 M sucrose (0.10 mg RAE g$^{-1}$ DW and 0.11 mg RAE g$^{-1}$ DW).

### 3.2. Pyrrolizidine Alkaloids Presence in Wild and In Vitro Growing Plants

Identification of pyrrolizidine alkaloids (PAs) in wild- and in vitro grown *R. umbellata* plants was performed by GC-MS analyses of obtained methanol extracts. In wild-growing plants, six PAs, namely, 7-angeloyl heliotridane, 7-angeloyl heliotridine, lindelofine, 7-angeloyl-9-(+)-trachelanthylheliotridine, punctanecine, and heliosupine were identified (Figure 3).

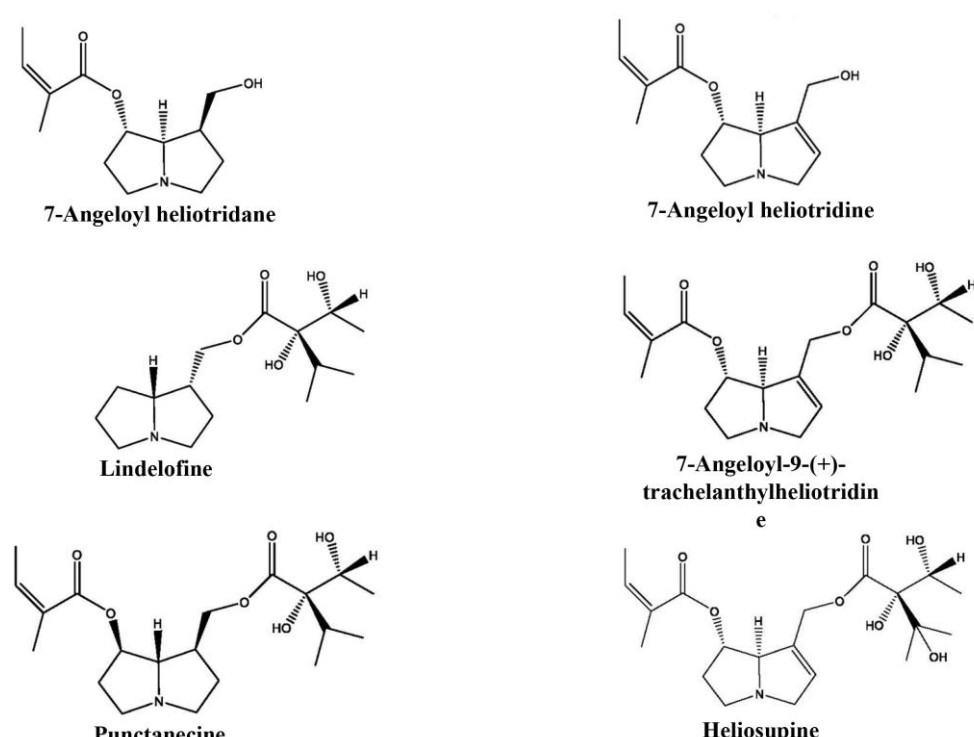

**Figure 3.** Structures of pyrrolizidine alkaloids detected in wild-growing *R. umbellata*.

The PAs were mostly detected in leaves and flowers, with the exception of heliosupine found only in leaves, and lindelofine detected in all tested plant parts (Table 3). Apart from the identified compounds, nine saturated and four unsaturated PAs of unknown structure were detected in leaves of wild-growing plants (data not shown). On the other hand, the presence and variability of PAs in in vitro plants grown under common conditions (0.1 M sucrose) were notably lower. Namely, only 7-angeloyl heliotridane and lindelofine were detected in leaves and roots, respectively.

**Table 3.** Pyrrolizidine alkaloids detected in samples of wild- and in vitro grown *R. umbellata* plants. Flw—flowers; lv—leaves; rt—roots.

| Pyrrolizidine Alkaloids | Retention Time (min) | Wild-Growing Plants | In Vitro Plants |
|---|---|---|---|
| 7-angeloyl heliotridane | 12.349 | flw, lv | lv |
| 7-angeloyl heliotridine | 13.171 | flw, lv | |
| lindelofine | 15.176 | flw, lv, rt | rt |
| 7-angeloyl-9-(+)-trachelanthylheliotridine | 19.571 | flw, lv | |
| punctanecine | 19.746 | flw, lv | |
| heliosupine | 20.288 | lv | |

### 3.3. Antioxidant Activity

In further work, antioxidant properties were evaluated for both wild-growing and in vitro cultivated *R. umbellata* plants. DPPH and ABTS assays were used and $IC_{50}$ values were determined (Table 4).

**Table 4.** Antioxidant activity of methanol extracts of *R. umbellata* plants from the natural habitat and grown in vitro. Results represent the mean values $\pm$ SE of the three repeated measurements. Different letters designate statistical difference at $p \leq 0.05$ level of significance.

| | $IC_{50}$ (mM RAE g$^{-1}$ DW) | |
|---|---|---|
| **Wild Growing Plants** | **DPPH$^\bullet$** | **ABTS$\bullet$+** |
| Leaves | 2.31 $\pm$ 0.23 *a* | 0.77 $\pm$ 0.01 *a* |
| Flowers | 4.14 $\pm$ 0.77 *a* | 1.15 $\pm$ 0.28 *a* |
| Roots | 1.71 $\pm$ 0.54 *a* | 1.03 $\pm$ 0.17 *a* |
| Stems | 52.56 $\pm$ 3.57 *b* | 10.06 $\pm$ 3.35 *b* |
| **Plants grown in vitro** | | |
| 0.3 M sucrose | 0.52 $\pm$ 0.04 *a* | 0.27 $\pm$ 0.01 *a* |
| 0.1 M sucrose | 2.07 $\pm$ 0.33 *b* | 1.06 $\pm$ 0.16 *b* |

The results obtained from extracts derived from wild-growing plants showed that all plant parts, except stems, had high antioxidant activity; the lowest $IC_{50}$ were determined for leaves in ABTS, and roots in the DPPH test (0.77 and 1.71 mM RAE g$^{-1}$ DW, respectively). Antioxidant capacity of the leaves was enhanced in the in vitro conditions, but depended on the applied sucrose concentrations (Table 4). $IC_{50}$ values pointed out that 0.3 M sucrose could be considered as the most appropriate one (0.52 and 0.27 mM RAE g$^{-1}$ DW in DPPH and ABTS assay, respectively). Comparison of the antioxidant activity with TPC, TFC, RA, and LAB values indicated that only TPC were significantly correlated to the antioxidant capacities of plant extracts, with $R^2 = 0.9351$ and $R^2 = 0.9384$, as determined by the DPPH and ABTS assay, respectively.

### 3.4. Cytotoxic Activity

The results obtained in the MTT assay showed that the most susceptible cell line was HT-29, where $IC_{50}$ values were determined for all tested extracts (leaves harvested from wild-growing plants and the ones grown in vitro on 0.3 M and 0.1 M sucrose, Table 5).

**Table 5.** Cytotoxic effect of *R. umbellata* leaves extracts towards selected cell lines. $IC_{50}$ values are expressed in mg mL$^{-1}$. SI was calculated as the ratio of $IC_{50}$ values determined for normal MRC-5 cells and for each cancer cell line. The value > 1 indicates higher toxicity against cancer cells, while the value $\geq$ 2 shows high selectivity to cancer cells.

| | HT-29 | | A549 | | MRC-5 |
|---|---|---|---|---|---|
| | IC50 (mg mL$^{-1}$) | SI | IC50 (mg mL$^{-1}$) | SI | IC50 (mg mL$^{-1}$) |
| **Wild-growing leaves** | $1.65 \pm 0.18$ | Na | Nd | Na | Nd |
| **In vitro grown leaves** | | | | | |
| 0.3 M sucrose | $1.25 \pm 0.14$ | 2.56 | $4.00 \pm 0.17$ | 0.80 | $3.20 \pm 0.15$ |
| 0.1 M sucrose | $3.19 \pm 0.11$ | 0.90 | $3.50 \pm 0.22$ | 0.80 | $2.90 \pm 0.20$ |

Nd—not determined in applied concentration range (up to 4 mg mL$^{-1}$); Na—not applicable, due to the fact that $IC_{50}$ value was not determined in tested concentration range, at least for one cell line.

Actually, the extract produced from wild-growing leaves was effective only against HT-29 cells. On the other hand, extracts of in vitro samples were cytotoxic against all three tested cell lines. The lowest $IC_{50}$ value was determined in the case of HT-29 cells treated with 0.3 M sucrose extract. Furthermore, the activity values of the extracts obtained from leaves harvested from 0.3 M sucrose and from nature were comparable ($IC_{50}$ values of 1.25 mg mL$^{-1}$ and 1.60 mg mL$^{-1}$, respectively), while 0.1 M sucrose showed a notably lower effect. In addition, the calculated selectivity indices (SIs), being used to compare activity on cancer and normal cells, indicated that the selectivity for cancer cells was observed in the case of 0.3 M sucrose extract and HT-29 cell line. Although SI was not calculable for the wild-growing plant extract, it can be noted that the selectivity against the HT-29 cell line was high, due to the lack of cytotoxicity on normal MRC-5 cells.

## 4. Discussion

Production of plant secondary metabolites using in vitro technology has a lot of advantages. It does not depend on the seasonal variations and/or the influence of the environmental factors, and additionally, enables the production of plant material without disturbing the environment. Growing in strictly controlled, optimized conditions could increase yield of some pharmaceutically important secondary metabolites, or even production of certain compounds that are usually not detected in plants from the natural habitats [25,26]. *R. umbellata*, which is considered as rare and endangered in Serbia, has already been successfully introduced in in vitro culture; moreover, the protocol for optimized growth and biomass production was established in our previous work [17]. However, until now, the comparative analysis of chemical composition and biological activities of the extracts from in vitro grown plants or plants from nature was lacking.

Concerning the PAs presence, it is worth noting that Ganos et al. [27] detected echinatine, echinatine N-oxide, and rinderine N-oxide in *R. graeca* samples collected from nature. In addition, previous investigations of our group on wild-growing *R. umbellata* showed that the extracts contained versatile PAs, including 7-angeloyl-9-(+)-trachelanthylheliotridine, lindelofine, punctanecine, 7-angeloyl heliotridane, 7-angeloyl heliotridine, heliosupine, lindelofine-N-oxide, heliosupine-N-oxide and 9-(+)-trachelanthyllaburnine, echinatine, and 7-angeloyl rinderine, but their presence was variable and depended on harvesting season, plant part, and extraction procedure [18,19]. Indeed, it can be assumed that relating to harvesting season, detected PAs content depends on different abiotic factors, such as temperature, light, and air humidity, as well as stages of plant development [28]. This is in line with the results obtained in this work for the methanol extracts of leaves, flowers, and roots of wild-growing *R. umbellata*. Among detected PAs, six were of identified structures: 7-angeloyl heliotridane, 7-angeloyl heliotridine, lindelofine, 7-angeloyl-9-(+)-trachelanthylheliotridine, punctanecine, and heliosupine. Taking into account that plant material and season of collection were not the same in previous studies [18,19] and this work, observed differences are to be expected. Since plant material in this work was col-

lected separately for different organs, one could note that the highest variability of PAs is present in the leaves and flowers.

On the other hand, *R. umbellata* grown in vitro in common conditions (0.1 M sucrose) produced small amounts of non-versatile PAs, namely, 7-angeloyl heliotridane and lindelofine. Scarce detection of PAs is in line with the literature data indicating that certain compounds, such as nicotine, some PAs, and tropane alkaloids were produced only in small amounts, or not produced at all, in in vitro culture [29]. In contrast, some literature data indicated that in vitro grown plants could also produce PAs in certain conditions; for example, Graikou et al. [30] detected echinatine, echinatine N-oxide, and rinderine N-oxide in seedlings and hairy roots of *R. graeca*. However, considering that some alkaloids are strongly toxic for mammals, inducing neurotoxic, hepatotoxic, mutagenic, and carcinogenic effects [28,31], in vitro plant culturing which could prevent or decrease alkaloids production is of great importance, especially in the case of medicinal plants. Accordingly, this method could be considered suitable for the production of some nontoxic, pharmaceutically important secondary metabolites of *R. umbellata*.

Phenolic profile analysis of wild-growing *R. umbellata* plants was performed for the first time in this study. The results showed that the distribution of TPC is the same in roots and flowers, while the highest content was measured in leaves samples. In addition, the TFC of different parts was determined in the following descending order: leaves > roots > flowers, while in stem it was determined only in traces. Furthermore, RA was detected in leaves and flowers, while LAB was determined only in roots. Similar results, with high values of TPC, TFC, and RA, but lack of LAB, were determined for aerial parts of *R. graeca* collected from nature [27].

Concerning determination of phenolic compounds in in vitro grown plants, it is worth noting that the previous research pointed out that the optimization of culturing conditions could increase their production [32,33]. Indeed, the level of secondary metabolites depends on numerous factors and among them are the availability of carbohydrates in the media, which used as a source of carbon and energy [32]. High concentrations of carbohydrates can affect cell metabolism and change osmotic potential, consequently inducing an increase in secondary metabolites production [32,34,35]. Considering this, we cultivated plants in the presence of variable amounts of different carbohydrates. Namely, sucrose, glucose, and fructose were added, all in the concentration range of 0.003–0.3 M. Their effect was variable, but comparison of all tested phenolics in wild growing and in vitro cultivated plants pointed out that TFC and RA content were notably increased in leaves of plants cultivated on media containing 0.3 M of sucrose. If this fact is taken into consideration together with the result showing that TPC in plants grown at 0.3 M sucrose is still high and comparable to the values determined in leaves collected from nature, it can be denoted as the most appropriate medium for the production of the phenolic compounds. However, according to our previous results [17], 0.3 M sucrose considerably reduced *R. umbellata* in in vitro growth and biomass production in comparison with optimal sucrose concentration (0.1 M). This is in line with previously published data that also indicate inverse proportionality of RA content and biomass production. This was confirmed for *Perilla frutescens* [36] and for *Salvia officinalis* [37]. A possible explanation for this phenomenon could be found in the fact that high concentrations of carbohydrates act as stressors and induce changes in cell metabolism and, consequently, inhibit plant growth but at the same time stimulate the production of secondary metabolites [5]. Consequently, a compromise between secondary metabolites content and biomass production should be made when the optimal conditions for in vitro production of bioactive substances are established. In accordance with that, both 0.1 M and 0.3 M sucrose were selected for further investigation of biological activities. The first one was selected due to the highest biomass yield, which was previously confirmed [17], but also due to the relatively high amount of all determined phenolic compounds. The second concentration was chosen as the one which provided the highest secondary metabolites production. The results of DPPH and ABTS assays performed with the extracts obtained from the wild-growing plants showed that moderate antioxidant activity was detected

for the ones prepared from roots, leaves, and flowers, while the antioxidant effect of stem extract was weak. Concerning plants grown in vitro, it is observable that antioxidant properties were the best for 0.3 M sucrose. If the values determined for TPC, TFC, RA, and LAB are taken into consideration, it seems that a notable increase in TPC contributed to elevated antioxidant activity. Actually, numerous literature data indicate that TPC, including flavonoids and RA, possess a strong antioxidative potential and could protect cells from oxidative stress and its possible consequences [38,39]. In accordance with the fact that oxidative stress is involved in numerous disorders, such as cardiovascular, neurological, respiratory, and kidney diseases, as well as diabetes mellitus, rheumatoid arthritis, cancer, aging, and aging-related diseases [40], potential applications of *R. umbellata* extracts could be numerous. Furthermore, to determine the cytotoxic potential of *R. umbellata*, comparative analysis of both wild-growing and in vitro cultivated plants was carried out. As for the antioxidant properties, the extracts of leaves obtained at 0.1 M and 0.3 M sucrose were used, but the extract of wild-growing plants' leaves was also involved. Efficacy was screened against two cancer cell lines, namely, lung (A549) and colorectal adenocarcinoma (HT-29). However, considering that the potential chemotherapeutics should possess stronger cytotoxicity against cancer than normal cells, normal lung fibroblasts were involved as well. Although the overall activity was weak, some observations could be made. If a comparison of the cell line susceptibility is made, one could note that HT-29 cells were highly sensitive, at least to leaves extracts of wild-growing and 0.3 M sucrose plants. Simultaneously, the lack of activity and notably lower activity on the normal MRC-5 cell line of the wild-growing and 0.3 M extracts, respectively, together with the literature data pointing out that SI $\geq$ 2 is an indication of high selectivity [24], showed that these extracts could be considered as effective against colorectal cancers. Certainly, high values of determined $IC_{50}$ against HT-29 indicated that crude methanol extracts could be denoted as a solid base for the identification of active cytotoxic principle/principles that are potentially applicable, at least against colorectal adenocarcinoma. On the other hand, lung carcinoma A549 cells were highly resistant to all tested extracts and, moreover, their sensitivity was even lower compared with normal MRC-5 cells. High resistance of this cell line could be attributed to the elevated level of catalase activity and glutathione, responsible for high intrinsic level of antioxidant protection, which is multifold higher in this cell line than in the normal lung fibroblasts [41].

Since the bioactive substances are responsible for the observed cytotoxicity, extract obtained from nature should be separately discussed from the ones prepared from in vitro grown leaves. When it comes to the leaves of wild-growing plants, both phenolics and PAs probably contributed to the observed activity, while in the case of the in vitro leaves extracts, the latter one should be omitted. Actually, phenolic compounds, although well known as potent antioxidants, in certain circumstances (high concentrations and/or the presence of metal ions) could induce a pro-oxidative effect [42]. Pro-oxidant agents could cause strong oxidative stress, which consequently could lead to cell death [43]. Furthermore, literature data show that RA and LAB also possess cytotoxic potential [14,15], indicating that they probably influenced the observed activity, especially in the case of the extracts of in vitro grown plants, since their levels were notably higher within them than within the extracts obtained from leaves from nature. In addition, PAs of the leaves collected from nature could also contribute to overall cytotoxic properties. This is supported by the fact that alkaloids, including PAs, are characterized by anticancer activity [44]. Mechanisms of their anticancer activity could be diverse, including the effect on tubulin polymerization, as it was shown in our previous work for lindelofine-N-oxide, isolated from *R. umbellata* [18].

A general view of the presented findings could be helpful for all researchers interested in ex situ conservation using in vitro techniques, and in utilization of biological activity of rare and endangered plant species. Considering possible use of *R. umbellata*, it is worth noting that its growing in in vitro conditions opens new possibilities for the production of biological active compounds. This would significantly reduce the pressure on the wild growing *R. umbellata* populations and prevent its further uncontrolled harvesting



from nature. In addition, in vitro techniques could enable *R. umbellata* reintroduction to natural habitats.

## 5. Conclusions

Chemical characterization revealed for the first time contents of total phenolics, flavonoids, rosmarinic, and lythospermic acids B in both, wild- and in vitro grown *R. umbellata*, as well as PAs presence in in vitro cultivated plants. Furthermore, this research pointed out that 0.1 M and 0.3 M sucrose in the growth media provided the highest yield of bioactive natural products with antioxidant properties. Although cytotoxic effect was mild and its selectivity was limited to colorectal HT-29 cell line, the obtained result encourages further study that should be directed to the determination of the active principals responsible for this activity, as well as to screening of numerous cancer cell lines.

**Author Contributions:** S.T., conceptualized the research, collected and statistically analyzed the data, and wrote and prepared the final version of the manuscript; M.P., performed the in vitro experiments, measured total phenolic and flavonoid content, and determined antioxidant activity; B.N., developed the concept and supervised research of the cytotoxicity experiments, and wrote and prepared the final version of the manuscript; B.M., performed GC-MS/FID analysis of PAs; S.C., performed the cytotoxicity experiments and the literature research; M.B., prepared extracts and tested biological activity; S.Ž., performed HPLC-DAD analysis of phenolic compounds. All authors have read and agreed to the published version of the manuscript.

**Funding:** This research was funded by the Ministry of Science, Technological Development and Innovation of the Republic of Serbia (grants: 451-03-47/2023-01/ 200007; 451-03-47/2023-01/ 200178).

**Data Availability Statement:** Raw data are available by request via email to slatod@ibiss.bg.ac.rs.

**Conflicts of Interest:** The authors declare no conflict of interest.

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
