# Peer review of "Chemical Characterization, Antioxidant Activity, and Cytotoxity of Wild-Growing and In Vitro Cultivated Rindera umbellata (Waldst. and Kit.) Bunge"

_horticulturae, doi:10.3390/horticulturae9030381_

Round 1

Reviewer 1 Report

Manuscript with the title “Chemical Characterization, Antioxidant Activity, and Cytotoxity of Wild-Growing and In Vitro Cultivated Rindera umbellata (Waldst. & Kit.) Bunge” provides first chemical characterization and biological activity, comparatively on wild-sourced vs in vitro plants, of a rare species growing in south-east Europe.

The research is thorough and detailed, results are well presented.  However, there is some disconnection or dissonance between introduction and aim of the research. I suppose the wild-harvested plants were used only for comparison for qualitative parameters of in vitro ones. Because it is less advisable to put in evidence the biological activity potential of an already rare and protected plant species, that could encourage people to source it for medicinal/para-medicinal purposes. Also, I suppose the appropriate permission was obtained prior to collecting the wild plants. If the in vitro plants can be a source of valuable compounds in a feasible way, then this is important and are great news.

I advise authors to integrate a paragraph in the discussion in order to highlight the potential applications they had in might for their study. Please do not omit to discuss about the measures to protect this species and how bio-conservation is also strongly considered.

Best regards.

Author Response

We thank  for all the suggestions that improve the quality of the manuscript. In addition, we respect the positive evaluation of our work. In order to meet the reviewer’s suggestions, we provide further corrections/improvements and answers:

  • Concerning some disconnection or dissonance between introduction and aim of the research, we have made some improvements in the Introduction section in order to meet this suggestion (Introduction Section, please, see lines 89-93).
  • Concerning the role of wild-harvested plants, we have used them together with in vitro ones in order to provide comparative analysis of chemical composition and biological activity. It is truth that determination of biologically active substances could encourage people to source the plant material, even in the case of rare and endangered plants. However, it is worth noting that plants grown in vitro could be used to reintroduce them in the nature, at least to growth them in a plantation. For that reason, we find that a comparative analysis provided in this research could be of special importance. In order to better elucidate this fact, we have made additional explanation in the Discussion section (Discussion Section, please, see lines 461-468).
  • Concerning the appropriate permission for collecting the wild plants, it has been provided. That is permission for collecting of strictly protected and protected plant species for scientific research purposes No. 353-01-184/2011-03. We have now included this information in the revised manuscript (Material and Methods Section, Paragraph 2.1. Wild-Growing Plant Material; please, see lines 97-99).
  • Concerning potential applications of our findings, we have added appropriate information in the Discussion Section. Since a notable antioxidant activity was observed, potential applications certainly include variable diseases with oxidative stress etiology (please see lines 418-421). In addition, cytotoxicity against colorectal adenocarcinoma cells is potentiated in a revised manuscript (please see lines 437-438).
  • Finally, in order to highlight the protection and bio-conservation measures of R. umbellata, we have made some upgrades of the Discussion Section (please, see lines 461-468).

Reviewer 2 Report

Material and methods

Why do you use only ABTS and DPPH test to probe antioxidant activity?

The use of A549 and MRC-5 cells is clear, however, HT-29 their use is not clear, please justify it.

About the cell culture, the authors evaluate viscosity, osmotic pressure, to evaluate culture conditions?

Section 3.3

Please delete next paragraph:

“Taking into account the high content of phenolic compounds in the extracts, some antioxidant potential was anticipated” and re-write next paragraph:

“it was further evaluated for both wild-growing 271 and in vitro cultivated R. umbellata plants. For that reason, DPPH and ABTS assays were 272 used and IC50 values were determined (Table 4).”

Line 295 – 297, Please add values of r2, to probe high correlation

Author Response

We thank for all the suggestions that improve the quality of the manuscript.

Why do you use only ABTS and DPPH test to probe antioxidant activity?
Answer:  Current manuscript is a part of a wider research on the R. umbellata and includes numerous analyses on both, phytochemical and physiological characterization of this endangered plant species. Since we have worked with a quite limited amount of plant material, we have chosen only these two most commonly used assays to gain a basic insight into the antioxidant activity of plant extracts.

The use of A549 and MRC-5 cells is clear, however, HT-29 their use is not clear, please justify it.

Answer: Searching for anticancer effect commonly starts with screening of cytotoxic properties against various cancer cell lines. Due to limited amount of test substances only two cancer cell lines were involved in this research: lung adenocarcinoma (A549) and colorectal adenocarcinoma (HT-29). In addition, in order to check whether the cytotoxic effect is selectively manifested only towards cancer cells, but not against normal ones, the normal human fetal lung fibroblasts (MRC-5 cells) were also included in the experimental set-up. Actually, a good anticancer agent should selectively kill only cancer, but not normal cells. By including the cell line of normal fibroblasts, we enabled calculation of selective indices and consequently determination of selective anticancer activity.

About the cell culture, the authors evaluate viscosity, osmotic pressure, to evaluate culture conditions?

Answer: It was stated in the manuscript that the procedure of cytotoxicity monitoring was exactly the same as in our previously published paper (Nikolić et al., 2015). Since in this previously published paper (Nikolić et al., 2015) we have stated that the used protocol was as the one provided in Hansen et al. (1989), we have changed the reference in the revised manuscript (cited Hansen et al., 1989, instead of  Nikolić et al., 2015). This methodology is routinely used by numerous authors, since it allows obtaining of reliable and repeatable results. In order to provide more detailed description of the used methodology, some upgrades were made in the Material and Methods Section (Paragraph 2.8 Cytotoxicity Assay, please, see lines 186-192). However, MTT assay does not provide for special estimation of viscosity and osmotic pressure. Special measuring of viscosity of culture media seems to be important for optimized computational fluid dynamics analysis of in vitro devices, but not for MTT assay. 

Hansen, M.B., Nielsen, S.E., Berg, K. Re-examination and further development of a precise and rapid dye method for measuring cell growth/cell kill, J Immunol. Methods 119 (1989) 203-210.

Section 3.3

Please delete next paragraph:“Taking into account the high content of phenolic compounds in the extracts, some antioxidant potential was anticipated” and re-write next  paragraph: “it was further evaluated for both wild-growing and in vitro cultivated R. umbellata plants. For that reason, DPPH and ABTS assays were 272 used and IC50 values were determined (Table 4).”

Answer: The suggested change has been accepted. Please, see lines 272-275.

Line 295 – 297, Please add values of r2, to probe high correlation.

Answer: We have now provided R2 coefficients and made appropriate changes in the revised manuscript. Please, see lines 297-300 in the Results, and lines 415-416 in Discussion paragraph.

Reviewer 3 Report

1. The subject addressed is one of interest, the aim was to comparatively analyze chemical composition and biological activity of wild- and in vitro grown Rindera umbellata.

2. The subject is original because explants were cultivated on 0.003-0.3 M sucrose, fructose or glucose. HPLC-DAD for quantifying rosmarinic (RA) and lithospermic B (LAB) acids and GC-MS/FID for qualitative pyrrolizidine alkaloids (PAs) detection were used. Antioxidant activity (DPPH and ABTS assays) and cytotoxicity (MTT test) were monitored.

I recommend a revision of the section: Discussions

Author Response

We thank for all the suggestions that improve the quality of the manuscript.

 I recommend a revision of the section: Discussions

Answer: We have improved the whole manuscript in accordance with the suggestions of two other reviewers, with emphasizing the Discussion Section.
